# Oscillatory Tracking of Continuous Attractor Neural Networks Account for Phase Precession and Procession of Hippocampal Place Cells

**Tianhao Chu**[1,*]
chutianhao@stu.pku.edu.cn

**Zilong Ji**[1,2,*]
zilong.ji@ucl.ac.uk

**Junfeng Zuo**[1]
zuojunfeng@pku.edu.cn

**Wen-Hao Zhang**[3]
wenhao.zhang@utsouthwestern.edu

**Tiejun Huang**[4]
tjhuang@pku.edu.cn

**Yuanyuan Mi**[5]
miyuanyuan0102@163.com

**Si Wu**[1,†]
siwu@pku.edu.cn

1, School of Psychology and Cognitive Sciences, IDG/McGovern Institute for Brain Research,
Peking-Tsinghua Center for Life Sciences, Academy for Advanced Interdisciplinary Studies,
Center of Quantitative Biology, Peking University.
2. Institute of Cognitive Neuroscience, University College London
3. Lyda Hill Department of Bioinformatics, O'Donnell Brain Institute, UT Southwestern Medical Center.
4. School of Computer Science, Peking University.
5. Center for Neurointelligence, School of Medicine, Chongqing University.
*: Equal contributions. †: Corresponding authors.

## Abstract

Hippocampal place cells of freely moving rodents display an intriguing temporal organization in their responses known as 'theta phase precession', in which individual neurons fire at progressively earlier phases in successive theta cycles as the animal traverses the place fields. Recent experimental studies found that in addition to phase precession, many place cells also exhibit accompanied phase procession, but the underlying neural mechanism remains unclear. Here, we propose a neural circuit model to elucidate the generation of both kinds of phase shift in place cells' firing. Specifically, we consider a continuous attractor neural network (CANN) with feedback inhibition, which is inspired by the reciprocal interaction between the hippocampus and the medial septum. The feedback inhibition induces intrinsic mobility of the CANN which competes with the extrinsic mobility arising from the external drive. Their interplay generates an oscillatory tracking state, that is, the network bump state (resembling the decoded virtual position of the animal) sweeps back and forth around the external moving input (resembling the physical position of the animal). We show that this oscillatory tracking naturally explains the forward and backward sweeps of the decoded position during the animal's locomotion. At the single neuron level, the forward and backward sweeps account for, respectively, theta phase precession and procession. Furthermore, by tuning the feedback inhibition strength, we also explain the emergence of bimodal cells and unimodal cells, with the former having co-existed phase precession and procession, and the latter having only significant phase precession. We hope that this study facilitates our understanding of hippocampal temporal coding and lays foundation for unveiling their computational functions.

36th Conference on Neural Information Processing Systems (NeurIPS 2022).

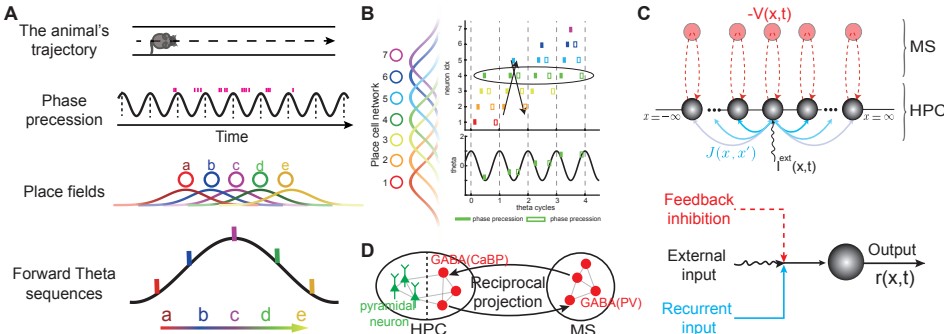

Figure 1: Theta phase coding and the continuous attractor neural network with feedback inhibition. A. Phase precession of a probe neuron and the theta sequence consisting of five neurons firing sequentially in a theta cycle. B. Both phase precession and procession of a probe neuron, and the corresponding forward theta sequence and backward theta sequence in a theta cycle. C. Up: the network model structure. Neurons in the CANN are arranged on an one-dimensional neuronal track, with each receiving feedback inhibition under the interaction with the medial septum. Below: input and output to a single neuron in the model. D. A schematic diagram of the reciprocal interaction between the hippocampus and the medial septum through GABAergic neurons.

# 1 Introduction

The mammalian hippocampus is responsible for spatial navigation [1, 2] and episodic memory [3–6]. Rodent studies have revealed two prominent firing features of hippocampal place cells during animals' locomotion. First, each place cell has localized spatial coding which fires in restricted regions of a given environment (place fields) [7, 8]. Second, the firing of a place cell displays detailed temporal organization known as 'theta phase precession', whereby neuronal spikes are elicited at successively earlier phases of the theta cycles of the local field potential (LFP) as the animal travels through the place field (Fig. 1A) [9, 10]. This phase precession of individual neurons results in that the activities of consecutively activated place cells along the animal's moving trajectory are compressed into a firing sequence within a theta cycle, named forward theta sequence (Fig. 1A) [10–14]. It has been hypothesized that such theta sequences compress sequential responses of neurons at the behavioral timescale (seconds) into the theta-cycle timescale (tens of milliseconds) in a temporally ordered way, which is short enough to enable spike-time-dependent plasticity (STDP) between neurons [15, 16]. Thus, theta phase precession may serve as a neural substrate for the formation of spatial and episodic memories [17].

Recently, experimental studies found that in addition to phase precession, many place cells in the hippocampus of freely moving rodents also exhibit phase procession, i.e., a place cell fire at progressively later phases in successive theta cycles as the animal traverses the place field (Fig.1B) [18–22]. Analogy to the formation of forward theta sequences, phase procession of place cells firing consecutively along the animal's moving trajectory results in reverse theta sequences. From the view of information processing, phase precession and procession correspond respectively, to the prospective and retrospective evaluations of animals' ongoing behaviors, which further supports the forward and reverse replays during memory consolidation [23–26]. Instead of existing alone in a theta cycle, a recent study showed that the reverse theta sequence co-exists with the forward theta sequence in individual theta cycles [27] (Fig. 1B). As it has far-reaching implications to brain functions, a large volume of modelling studies has been devoted to understand the generation of theta phase precession of place cells. These models can be roughly divided into two categories, with one focusing on the mechanism of single cell oscillation [9, 28–31] and the other on the mechanism of recurrent interactions between neurons [32–34] (see more discussions in Sec. 5). However, these models have not accommodated the newly found phase procession which co-exists with phase precession in theta cycles, and hence need to be re-evaluated.

In the current study, we propose a neural circuit model to elucidate the underlying mechanism of the co-occurrence of phase precession and procession in individual theta cycles, as well as many other recent experimental findings. Specifically, we consider a continuous attractor neural network (CANN) with feedback inhibition (Fig. 1C). The CANN is a conceptualized circuit model

for the network of place cells [35–38], and the feedback inhibition models the interplay between the hippocampus and the medial septum with reciprocally connected GABAergic neurons [39–42] (Fig. 1D). Without feedback inhibition, the CANN holds a localized bump-shaped activity state (resembling the decoded virtual position of the animal), which can smoothly track the movement of the external input (resembling the physical position of the animal). When a strong enough feedback inhibition is added, it destabilizes the neuronal firing and induces intrinsic mobility of the bump activity. This intrinsic mobility competes with the extrinsic mobility (arising from the external drive) and causes an oscillatory tracking state, i.e., the network bump sweeps back and forth around the external moving input. Intriguingly, we find that this oscillatory tracking naturally gives rise to the phase shift of place cell firing during animals' locomotion. At the neuron ensemble level, the forward and backward bump sweeps account for, respectively, the forward and reverse theta sequences of the place cell ensemble. At the single neuron level, individual neurons in the CANN exhibit both phase precession and procession as the external input travels through their firing fields, with precession and procession occurring in the forward and backward bump sweeps, respectively. Moreover, we find that the feedback inhibition strength is the key for generating different phase shift patterns in place cell firing. With a weak feedback inhibition, all the neurons in the CANN are bimodal cells, which possess co-existed phase precession and procession in their firings. With a strong feedback inhibition, all the neurons in the CANN are unimodal cells, which possess only significant phase precession in their firings. We also theoretically elucidate the underlying network dynamics for producing these firing behaviours. We hope that this study helps us to unveil the underlying mechanism of phase shifts of place cells, and sheds light on our understanding of temporal coding of hippocampal neurons and the related functions in spatial navigation and episodic memory.

## 2 The Computational Model

### 2.1 A CANN with feedback inhibition

For simplicity, we study a one-dimensional (1D) CANN (modeling animals' free moving on linear tracks), and results can be naturally generalized to the 2D case. The 1D CANN is a conceptualized circuit model of the place cell ensemble, which consists of neurons arranged conceptually on a 1-dimensional neuronal track according to their relative firing locations (Fig. 1C). It is noteworthy that the relative location of two neurons on the neuronal track is irrelevant with their relative anatomical location in the hippocampus, i.e., neurons situated in the CANN can be viewed as place cells being re-arranged according to their place fields during animals' locomotion on the linear track. Denote $U(x, t)$ as the synaptic input received by the neuron at location $x$, with $x \in (-\infty, \infty)$, and $r(x, t)$ the corresponding firing rate. The dynamics of the CANN is written as,

$$\tau \frac{dU(x, t)}{dt} = -U(x, t) + \rho \int_{-\infty}^{\infty} J(x, x') r(x', t) \, dx' - V(x, t) + I^{ext}(x, t), \quad (1)$$

$$r(x, t) = \frac{g U(x, t)^2}{1 + k\rho \int_{-\infty}^{\infty} U(x', t)^2 dx'}, \quad (2)$$

where $\tau$ is the neuron time constant, and $\rho$ represents the neuron density. The recurrent connections are translation-invariant, which is given by $J(x, x') = J_0/(2\pi a) \exp\left[-(x - x')^2/(2a^2)\right]$, with $J_0$ the maximum connection strength and $a$ the range of neuronal interactions. $I^{ext}(x, t)$ is the external input to the CANN, which resembles the physical location of the animal moving on the linear track. The nonlinear relationship between the firing rate $r(x, t)$ and the synaptic input $U(x, t)$ is implemented by the divisive normalization, whose strength is controlled by $k$ in the denominator of Eq.2, and $g$ is a gain factor. The divisive normalization implicitly models the contribution of inhibitory neurons to the pyramidal neurons, which could be implemented by shunting inhibition in the real neural system [43]. It has been shown that with $k < \rho J_0^2/(8\sqrt{2\pi}a)$, the CANN holds a continuous family of Gaussian-shaped stationary states (bump states), and these bumps form a manifold on which the network is neutrally stable [44, 45] (how the network bump state tracks the external input will be discussed in Sec. 2.2).

$V(x, t)$ in Eq. 1 represents the feedback inhibition received by each neuron at location $x$, whose dynamics is written as,

$$\tau_v \frac{dV(x, t)}{dt} = -V(x, t) + mU(x, t), \quad (3)$$

where $\tau_v$ is the time constant of $V(x, t)$, and $\tau_v \gg \tau$ holds, implying that the feedback inhibition is a much slower process compared to the neuronal response. The parameter $m$ controls the strength of feedback inhibition, i.e., the larger the $m$, the stronger the inhibition. It is noteworthy that the feedback inhibition described in Eq. 3 does not simply mean that neurons in the medial septum receive direct excitatory input from neurons in the CANN and in turn inhibit them accordingly, but instead, represents an overall effect of the interplay between the hippocampus and the medial septum. This interplay may be implemented by the long-range reciprocal projections via GABAergic neurons in the hippocampo-septo-hippocampo loop [40, 46–48] (see more discussion in Sec. 5).

## 2.2 Tracking behaviors of the CANN with feedback inhibition

In this section, we show how the network bump (internal representation) tracks the external input (external physical location). Without loss of generality, we consider that the external input to the CANN has the following Gaussian form:

$$I^{ext}(x, t) = \alpha \exp \left[ -\frac{(x - v_{ext}t)^2}{4a^2} \right], \tag{4}$$

where $v_{ext}$ represents the moving speed of the external bump input (resembling the moving speed of the animal on the linear track), and $\alpha$ controls the strength of the external input. It is known that without the external input drive, i.e., $\alpha = 0$, the network bump can move spontaneously due to the destabilization of neural firing by the feedback inhibition (Fig. 2A). Specifically, when $m > \tau/\tau_v$, the bump moves with an speed $v_{int} = (2a/\tau_v)\sqrt{m\tau_v/\tau - \sqrt{m\tau_v/\tau}}$ [49] (see Fig. 2B and SI.2 for the mathematical derivation). We term this spontaneously moving as the intrinsic mobility hereafter, which is independent of the external drive.

When both the external input (from sensory drive) and the feedback inhibition (from the interaction with the medial septum) are applied to the CANN, the bump mobility is determined by these two factors in a competitive way, that is, the external input tends to drive the bump to move at the speed $v_{ext}$, and the feedback inhibition tends to drive the bump to move at the intrinsic speed $v_{int}$. Depending on the relative strengths of these two factors (controlled by $\alpha$ and $m$ in Eq. 4 and Eq. 3,

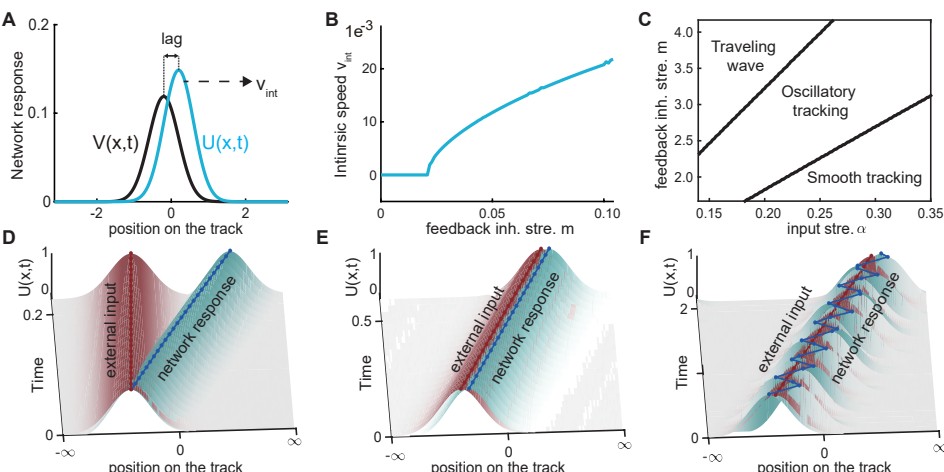

Figure 2: Tracking behaviors of the CANN with feedback inhibition. A. Without external input, the network bump moves spontaneously with a intrinsic speed $v_{int}$ due to destabilization from the feedback inhibition. The bump in blue represents $U(x, t)$ and the bump in black represents $V(x, t)$ which always lags behind the bump $U(x, t)$. B. Intrinsic speed $v_{int}$ as a function of the feedback inhibition strength $m$. C. The phase diagram of the tracking states with respect to the input strength $\alpha$ and the feedback inhibition strength $m$. D. Traveling wave state with a large $m$. E. Smoothing tracking state with a small $m$. F. Oscillatory tracking state with a moderate $m$. D-F: Bumps in red represent the external bump input (physical position of the animal), and bumps in blue represent the network bump (virtual position of the animal). For parameter settings of producing these figures, see SI.1 and the uploaded code.

respectively), the network exhibits three different tracking behaviors as summarized in Fig. 2C. Specifically, we can study the tracking behaviors by fixing one parameter while varying the other. While fixing the external input strength $\alpha$ and varying the feedback inhibition strength $m$, we have

- **Travelling wave**: when the feedback inhibition is relatively strong (large $m$), the network holds the travelling wave state (Fig. 2D), in term of that the bump moves spontaneously at the speed of $v_{int}$, independent of the external input. This is understandable: although the external input tends to attract the network bump, the feedback inhibition has a stronger force to push it away, which causes the bump to move with its intrinsic speed.

- **Smooth tracking**: when the feedback inhibition is relatively weak (small $m$), the network holds the state of smooth tracking (Fig. 2E), in term of that the bump is completely attracted by the external input and moves at the speed $v_{ext}$. This is because the feedback inhibition is not strong enough to push the network bump away.

- **Oscillatory tracking**: when the feedback inhibition is moderate, the network holds the state of oscillatory tracking (Fig. 2F), in term of that the overall motion of the bump follows the external moving input, while its position oscillates around the instant position of the external input. This oscillatory tracking originates from the competition between the intrinsic mobility caused by the feedback inhibition and the drive of the external moving input, which we explain in details below.

### 2.3 The oscillatory tracking state

In this section, we present the analytical solution of the network model in the parameter regime of oscillatory tracking (see SI.3 for mathematical details). Due to the Gaussian-like neuronal connections and global inhibition (Eq. 2), we find that the network states in Eq. 1- 3 can be approximated as Gaussian forms, which are expressed as:

$$\overline{U}(x,t) = A_u(t) \exp\left\{ -\frac{[x - z(t)]^2}{4a^2} \right\}, \tag{5}$$

$$\overline{r}(x,t) = A_r(t) \exp\left\{ -\frac{[x - z(t)]^2}{2a^2} \right\}, \tag{6}$$

$$\overline{V}(x,t) = A_v(t) \exp\left\{ -\frac{[x - z(t) + d(t)]^2}{4a^2} \right\}. \tag{7}$$

Here $A_u(t)$, $A_r(t)$ and $A_v(t)$ denote the bump heights at time $t$ (note that $U(x,t)$, $r(x,t)$ and $V(x,t)$ all have bump-shaped profiles). $z(t)$ denotes the position of bump centers of $U(x,t)$ and $r(x,t)$, which are always phase-locked on the neuronal track. $d(t)$ is the displacement of the bump centers between $U(x,t)$ and $V(x,t)$, due to the delayed feedback inhibition with respect to the neuronal

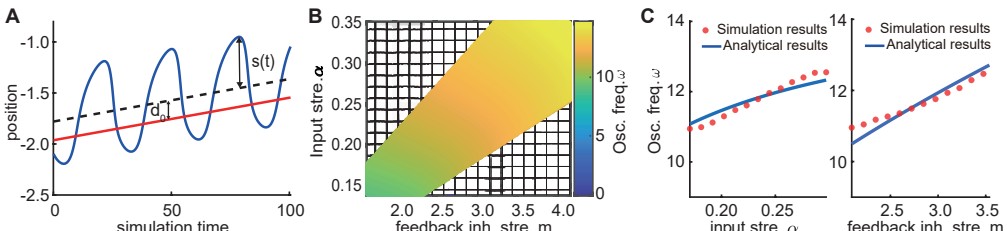

Figure 3: Oscillatory tracking in our model. A. Illustration of the oscillatory tracking as a sinusoidal wave with an positive offset $d_0$ around the external input. For simplicity, only the bump centers are shown (blue: network bump center; red: external bump center). B. The phase diagram of the oscillation frequency with respect to the input strength $\alpha$ and the feedback inhibition strength $m$. Grided area represents the traveling wave or the smooth tracking state (as shown in Fig. 2C) where no oscillation exists. C. Simulation (red dots) and analytical (blue lines) results of the oscillation frequency as a function of the input strength $\alpha$ and the feedback inhibition strength $m$, respectively. For parameter settings of producing these figures, see SI.1 and the uploaded code.

response in the CANN (see Fig. 2A). For studying the bump oscillation, we focus on the dynamics of $z(t)$, as the bump-shaped profiles remain unchanged across time. From the simulation (Fig. 2F), we assume $z(t)$ can be expressed as:

$$z(t) = s(t) + v_{ext}t, \quad \text{with} \quad s(t) = c_0 \sin(\omega t) + d_0. \tag{8}$$

Here, $s(t)$ denotes the displacement between the centers of the network bump and the external bump at time $t$ (Fig. 3A), which oscillates as a sine wave. $c_0$ and $\omega$ are positive values, denoting, respectively, the amplitude and the frequency of oscillation. $d_0 > 0$ is a constant representing the positive offset of bump oscillation with respect to the trajectory of the external input, which is observed in the experiment (see Fig.1 in SI.4). To obtain the parameter values $c_0$, $\omega$ and $d_0$ in Eq. 8, we can simplify the analysis by transforming the network dynamics into a low-dimensional space [45, 50]. This can be done by projecting the network dynamics (after substituting Eq. 5-7 into Eq. 1- 3) onto the motion modes that dominate the dynamics of the CANN, i.e., the bump position on the attractor manifold and the bump height (see SI.2 & 3 for the detailed derivation). The oscillation frequency is calculated to be (for other parameter values, see SI.1),

$$\omega = \sqrt{\frac{2\sqrt{\pi}\alpha a k (1+m)}{\tau\tau_v(J_0 + 2\sqrt{\pi}ak\alpha)}}. \tag{9}$$

Notably, the oscillation frequency $\omega$ scales sublinearly with both the strength of the external input $\alpha$ and the feedback inhibition $m$ (see the phase diagram in Fig. 3B). Increasing the strength of external input or the feedback inhibition enables the network bump to oscillate more frequently, and the simulation results agrees well with the theoretical analysis (Fig. 3C). Additionally, the expression in Eq. 9 depends on other parameters such as the neuronal interaction range $a$ and the global inhibition strength $k$, which allows us to tune the oscillation frequency into the range of theta band. We show below that this oscillatory tracking behavior naturally accounts for the forward and reverse theta sequences discovered in the hippocampus.

## 3 Oscillatory tracking accounts for the co-occurrence of phase precession and procession

Early rodents studies widely found the canonical forward theta sequences in animals' locomotion [10, 13, 14, 51], where the decoded position from the place cell population activity sweeps in position starting slightly behind the animal and ending ahead of it in individual theta cycles. A recent study re-examined this finding with a large cell population on a fine time scale, and found that the decoded virtual position rarely progresses exclusively in a single direction, but instead, progresses in both directions, with one traveling ahead of the animal and the other moving backward in the reverse

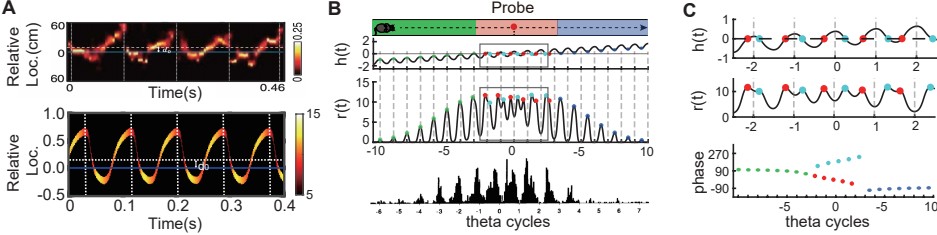

Figure 4: Oscillatory tracking accounts for the co-occurrence of phase precession and procession. A. Forward and backward theta sweeps in the experiment [27] (upper panel) and in our model (lower panel). B. First panel: three stages of the external input traversing the firing field of the probe neuron (green: entering, red: centering, blue: leaving); second panel: the displacement between the bump center and the probe neuron versus theta cycles; third panel: firing of the probe neuron versus theta cycles (red dots in the box: precession; blue dots in the box: procession); fourth panel: firing of a neuron as the animal moving through the place field (adapted from the experimental data [10]). C. The enlarged view of phase precession and procession in the box in B (upper two panels), and the phases of the firing peak of the probe neuron in individual theta cycles (third panel). For parameter settings of producing these figures, see SI.1 and the uploaded code.

direction of the movement (Fig. 4A upper panel) [27]. We show that the co-occurrence of forward and backward sweeps of the decoded position in individual theta cycles is naturally given by the oscillatory tracking behavior in our model. In our model, the external bump input represents the physical location of the rat on the linear track, and the network bump represents the decoded location from the place cell population activity. In each theta cycle, the network bump sweeps around the external bump in the forward and backward direction alternatively, which is analogy to the forward and reverse theta sequences found in the experiment (Fig. 4A lower panel). Additionally, our model predicts a positive offset $d_0 > 0$ in oscillatory tracking, as indicated by the experiment [27] (see detailed description in SI.4).

Below we show that the forward and backward sweeps during the oscillatory tracking in the CANN account for, respectively, the phase precession and procession of individual place cells. Without loss of generality, we study the firing phases of the neuron located at $x = 0$ (referred to as the probe neuron). Replacing $z(t)$ in Eq. 6 with the expression in Eq. 8, we obtain the firing response of the probe neuron at time $t$ as

$$r_0(t) = A_r(t) \exp\left[-\frac{(v_{ext}t + c_0 \sin \omega t + d_0)^2}{2a^2}\right] = A_r(t) \exp\left[-\frac{h(t)^2}{2a^2}\right], \qquad (10)$$

where $h(t) = v_{ext}t + c_0 \sin \omega t + d_0$ represents the displacement between the network bump center and the location of the probe neuron (Fig. 4B second panel). In the following discussion, we will refer to the time as the external bump reaches the location of the probe neuron as $t = 0$, with $t < 0$ representing the external bump moving on the left-side of the probe neuron and $t > 0$ representing the bump moving on the right-side of the probe neuron.

For the convenience of description, we first consider the case that the bump height $A_r(t)$ has little variation during the bump oscillation (for other situations, see Sec. 4). In such case, variations of the firing rate of the probe neuron is determined by the variations in $h(t)$, which is composed of a funnel-shaped signal $v_{ext}t$, and a sinusoidal signal $c_0 \sin \omega t + d_0$ whose amplitude is far less than the firing field of the probe neuron ($c_0 \ll a$, see SI.3 for the derivation of $c_0$). As the oscillation frequency is also much quicker than the moving speed, the sin wave can be seen as a signal being superposed on the funnel-shaped signal. This naturally gives the envelope of the firing rate to be of the waxing-and-waning shape, which is observed in the experimental data as the animal traversed the firing field of a place cell (Fig.4B bottom) [9, 10].

We further study the firing phase of the probe neuron in each theta cycle, which is reflected by the moment when the neuronal response reaches a peak in a theta cycle. From Eq.(10), we see that this peak firing rate is achieved when the absolute value of the displacement $|h(t)|$ reached the minimum value in each theta cycle. We distinguish three scenarios of the firing phase below (Fig.4B&C):

- **Fixed entry phase**. When the animal just enters the place field of the probe neuron, the location of the probe neuron is far away from the bump center, i.e., $h(t) < 0$ always holds. The minimum value of $|h(t)|$ in a theta cycle is reached when $c_0 \sin(\omega t) = c_0$. This gives the entry phase as a fixed value of $\phi = \omega t = \pi/2$.

- **Phase precession and procession**. As the animal approaches the center of the place field, i.e., in the range that $-c_0 - d_0 < v_{ext}t < c_0 - d_0$, the firing peak of the probe neuron is achieved when $h(t) = 0$. This gives two solutions of the phase for peak firing as follows:

$$\phi_f = -\arcsin\left[\frac{d_0 + v_{ext}t}{c_0}\right], \quad \phi_r = \pi + \arcsin\left[\frac{d_0 + v_{ext}t}{c_0}\right], \qquad (11)$$

  with $\phi_f$ occurring in the forward propagation of the network bump (forward theta sequences), and $\phi_r$ occurring in the reverse propagation of the network bump (reverse theta sequences) (Fig.4C upper two panels). As the animal travels through the place field of the probe neuron, $v_{ext}t$ increases from $(-c_0 - d_0)$ to $(c_0 - d_0)$. This leads to $\phi_f$ decreasing from $\pi/2$ to $-\pi/2$, and $\phi_r$ increasing from $\pi/2$ to $3\pi/2$, resulting in phase precession and procession, respectively.

- **Fixed departure phase**. When the animal is about to leave the place field of the probe neuron, $h(t) > 0$ always holds. The minimum value of $|h(t)|$ is reached in a theta cycle when $c_0 \sin(\omega t) = -c_0$, and this gives the departure phase as a fixed value of $\phi = \omega t = -\pi/2$ (Fig.4C bottom panel).

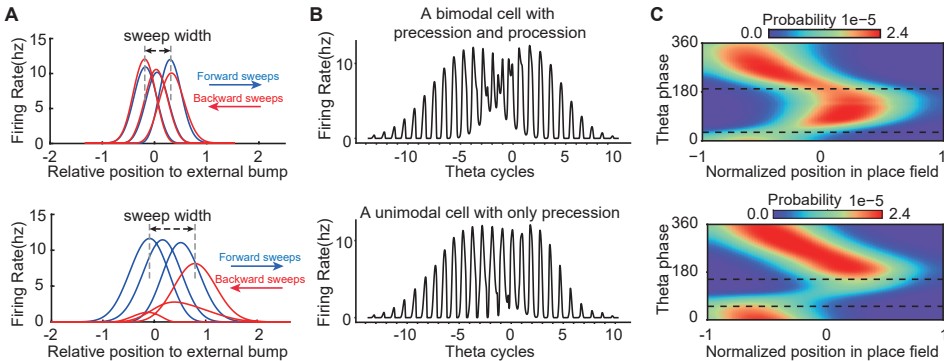

Figure 5: Emergence of bimodal cells and unimodal cells. A. Illustration of the variation of bump height and sweep width during oscillation. Upper panel: small $m$; lower panel: large $m$. B. Firing of a probe neuron versus theta cycles when the external input traverses the firing field. Upper panel: a simulated bimodal neuron with significant phase precession and procession; lower panel: a simulated unimodal neuron with only phase precession. C. Heatmaps of the averaged peak firing phase of all CANN neurons versus the normalized position in the corresponding firing field (-1 is entering the field, 1 is leaving). Upper panel: the averaged heatmap of bimodal neurons (simulated with a small $m$); lower panel: the averaged heatmap of unimodal neurons (simulated with a large $m$). For parameter settings of producing these figures, see SI.1 and the uploaded code.

Combing the above results, we see that as the animal traverses the place field, the probe neuron exhibits both phase precession and procession, and they occur in the forward and reverse theta sequences, respectively (i.e., in the forward and reverse propagation of the bump oscillation, respectively). Notably, our model also predicts a fixed entry phase as pointed out by the experimental data [9, 10] (note that the exact value of the entry phase is not important, as it depends on the reference point).

## 4 Feedback inhibition strength accounts for the emergence of bimodal cells and unimodal cells

In the above analysis, we have shown that under the interplay between external drive and feedback inhibition, neurons in the CANN exhibit both significant phase precession and procession in individual theta cycles, with the former determined by the peak firing time in the forward bump propagation and the later determined by the peak firing time in the backward bump propagation. Place cells in the hippocampus having this property are termed as bimodal cells [27]. In fact, previous experiments have already identified another type of place cells which only exhibit phase precession when the animal moving through the corresponding firing field [9, 10], and they are termed as unimodal cells. From the view of population decoding, these unimodal cells contain only forward sweeps of decoded position. There are compelling evidences that these two types of place cells participate differently in the encoding of spatial information. While the unimodal cells may exclusively represent possible future outcomes, the bimodal cells could also represent prior behavior in the reverse order. Here we show that by varying the feedback inhibition strength, our model can explain the emergence of both types of place cells discovered in the experiments.

For the simplicity of analysis in Sec. 3, we assumed that the bump height $A_r(t)$ (also, $A_U(t)$) has little variation during bump oscillation. In practice, this is only true when the feedback strength $m$ is relatively small. From the solution of the oscillation amplitude $c_0$ (see eqn. (70) in SI.3), we see that $c_0$ scales with $m$ in a sublinear way. When $m$ is small, the intrinsic mobility is small, and the bump is more anchored by the external drive than the intrinsic mobility, leading to a small oscillation amplitude. Because the height of the external bump is not changing across time, the network bump will accordingly have little variation in the height (Fig. 5A upper panel). Therefore, neurons in the CANN have both significant firing peaks in forward sweeps and backward sweeps (following the analysis in Sec. 3), and exhibit the property of bimodal cells. While, when $m$ is large, the intrinsic mobility becomes large, and the bump is less anchored by the external drive, leading to a large oscillation amplitude. As the internal bump oscillates to the location far from the external bump, the bump height starts to attenuate due to the weak external drive at that location. Since the feedback

inhibition is a much slower process compared to the neuronal response ($\tau_v \gg \tau$), the bump height will keep attenuating till it sweeps backward to the location of the external bump (Fig. 5A lower panel, also see the video demonstration in SI). This backward attenuation causes the firing peak during the backward sweep no longer significant, and hence the probe neuron appear to elicit only a single firing peak during the forward sweep in individual theta cycles (Fig. 5B), exhibiting the property of unimodal cells.

We examine the phase shift behaviors of neurons in the network by varying the feedback inhibition strength. To compare our simulation results with the experimental data, we follow the same process. The simulation results show that: 1) when $m$ is relatively small ($m = 3.125$, see SI.1 for all parameter settings), all neurons in the CANN are bimodal cells, which exhibit both significant phase precession and procession (Fig. 5C upper panel); 2) when $m$ is relatively large ($m = 3.15$), all neurons in the CANN are unimodal cells, which only exhibit significant phase precession (Fig. 5C lower panel). Notably, unimodal cells in our model also display a weak effect of phase procession (although bump height attenuates, they can still generate minor peak firings sometimes), although it is not as significant as that in bimodal cells. This is consistent with the experimental findings (see Fig.2 in SI.4), indicating that phase procession is a general property of hippocampal place cells [27].

## 5 Conclusions and Discussions

In the present study, we have proposed a continuous attractor neural network with feedback inhibition to elucidate the generation of theta phase shift of hippocampal place cells during animals' locomotion. We show that the interplay between the intrinsic mobility (arising from the feedback inhibition) and the extrinsic mobility (arising from the external drive) leads to the oscillatory tracking behavior of the network, which naturally accounts for the forward and backward sweeps of the decoded position (around the physical position) when the animal moves freely on a linear track. At the individual neuron level, the forward and backward sweeps of the network bump account for, respectively, the theta phase precession and procession of individual place cells. We also show that by tuning the feedback inhibition strength, neurons in the network can exhibit the property of either bimodal or unimodal cells, with the former having co-existed phase precession and procession in individual theta cycles, and the latter having only significant phase precession. Oscillatory tracking in attractor networks has been studied previously [52–54], but these works have not explored its biological implications. To our knowledge, our work is the first one that formally links oscillatory tracking to phase shift of place cells.

Previous modeling works have mainly focused on the generation of theta phase precession in place cell firing, and these models can be divided into two categories, with one focusing on the mechanism of single cell oscillation [9, 28–31] and the other on the mechanism of recurrent interactions between neurons [32–34]. A representative model of the former is the dual oscillator model [9, 30], which produces phase precession via the interaction between two oscillatory signals of different frequency. One signal represents the baseline somatic oscillation and the other with a slightly higher frequency represents the dendritic oscillation, and the superposition of two signals leads to that the firing peak of the cell progresses in successive theta cycles. A representative model of the latter is the spreading activation model [32], which produces phase precession as a result of the asymmetric propagation of neural activity along the motion trajectory of the animal, caused by asymmetric synaptic interactions. It is noteworthy that these two different kinds of models need not to be mutually exclusive, as each of them explains some aspects of the phase precession behavior [55]. Comparing with previous works, our model has two novel contributions: 1) explaining the co-existence of phase precession and procession, while previous models explained only phase precession; 2) elucidating the mechanism (via tuning the feedback inhibition strength) of generating bimodal and unimodal cells, which was not considered in previous models. Moreover, we carried out comprehensive analysis of the network dynamics to unveil the underlying neural mechanisms clearly.

The feedback inhibition plays an indispensable role in our model. It should be emphasized that the simplified feedback inhibition form in our model does not mean a direct reciprocal connection between the pyramidal neurons (place cells) in the hippocampus (HPC) and the GABAergic neurons in the medial septum (MS). In fact, there is not direct projection between these two neuron populations [40, 41, 56, 57]. Instead, the only feedback pathway between the HPC and the MS arises from reciprocal projections between the GABAergic (CaBP-containing) neurons in the HPC and the GABAergic (PV-containing) neurons in the MS [40]. Since the GABAergic (CaBP-containing) neurons receive

convergent excitatory inputs from the pyramidal neurons (place cells) in the HPC, a possible way of realizing the effect of feedback inhibition could arise from the complex interactions among the pyramidal neurons and GABAergic (CaBP-containing) neurons in the HPC, and the GABAergic (PV-containing) neurons in the MS. Moreover, the feedback inhibition may have layer-specific effects on neurons in the HPC pyramidal layer. Since bimodal cells have firing patterns similar to neurons in the deep HPC pyramidal layer while unimodal cells are more similar to neurons in the superficial layer [27, 58], we hypothesize that neurons in the deep HPC pyramidal layer may receive a larger feedback inhibition effect than neurons in the superficial layer, leading to the differentiation of unimodal and bimodal cells. In the future work, we will extend the current model to include more detailed biological structures accounting for the realizations of feedback inhibition and other properties in the CANN.

## Acknowledgement

This work was supported by Science and Technology Innovation 2030-Brain Science and Brain-inspired Intelligence Project (No.2021ZD0200204, No. 2021ZD0203700 / 2021ZD0203705, Y.Y. Mi), Guangdong Province with Grant (No.2018B030338001), the National Natural Science Foundation of China (No.4861425025, T.J.Huang, N0: T2122016, Y.Y.Mi), the Fundamental Research Funds for the Central Universities (No.2022CDJKYJH034), and Beijing Academy of Artificial Intelligence

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
