# OpenReview forum: "Oscillatory Tracking of Continuous Attractor Neural Networks Account for Phase Precession and Procession of Hippocampal Place Cells"
_NeurIPS.cc/2022/Conference — NeurIPS 2022 Accept_

### Official Review · Reviewer_Q2BM · 2022-07-08

**Rating:** 6
**Confidence:** 4
**Soundness:** 3 good
**Presentation:** 3 good
**Contribution:** 3 good

**Summary:**

This paper proposed a CANN model with feedback inhibition to elucidate the mechanism underlying phase precession and procession of the firing of hippocampal place cells. Compared to standard CANN models, a feedback inhibition mechanism is used to perturb the steady state of a standard CANN, so that the activity bump oscillates around the true location of a rat. This paper derived analytic formulas of the CANN under different input and feedback conditions. This paper derived the expression of the trajectory of the oscillatory state. This paper simulated the proposed model and observed that the activity bump sweeps back and forth across the true location, similar to what observed in previous experiments. The firing rate dynamics of the model show behaviors similar to phase precession and procession.

**Questions:**

1. 1D model can only explain the phase precession and procession of a rat running on a linear track. How the proposed model can be generalized to 2D cases where a rat can run in an open field?
2. The synaptic strength of the CANN is preconfigured through an ideal Gaussian form. In reality, such synaptic strengths might fluctuate continuously due to plasticity. Does the main results still hold if synaptic strengths become somewhat not ideal? How the synaptic strengths are learned through experiences?
3. Is there a missing link between the firing rate model of a CANN and precise spike firing times observed during phase precession and procession?

**Limitations:**

Yes.

**Strengths And Weaknesses:**

Strengths:
+ The presentation and writing of this paper is good.
+ The mechanism underlying phase precession and procession is still unknown. It is meaningful to propose computational models to understand the mechanism.
+ The theoretical results are sound and comprehensively show the steady state behaviors of the proposed model.
+ Both analysis and simulations show that the proposed model provides a reasonable and relatively abstract explanation of both the phase precession and procession phenomena.

Weaknesses:
- The CANN model with feedback inhibition is itself an existing model of oscillatory behaviors of activity bumps. The main novelty is the analysis and applications of such models to understand the phase precession and procession.
- The animal movement model is too simple (movement along a line) to account for many more realistic movement scenarios.
- The synaptic strength distribution of the model is assumed to be an idealized form without considering learning effects.
- Phase precession and procession are defined in terms of spiking times during consecutive  theta cycles. The previous model does not simulate the generation of individual spikes.

Update after rebuttal:
Thank you to authors for their responses. From their responses, the major difficulty of generalizing the current framework lies in the lack of experimental records. Regardless, it would be helpful to generalize their framework or experimental records to 2D or learning scenarios. Such generalization will increase the impact of this paper and might be very helpful to machine learning community.

---

> ### Author Response · Authors · 2022-08-01
> **Response to Reviewer Q2BM:**
>
> **We acknowledge the valuable and encouraging comments of the reviewer and would like to address them in the below.**
>
> **About the 2D Case:**
>
> 1) First of all, we would like to point out that under the 2D case of modeling the animals’ moving behavior in an open field, the mechanism for generating theta phase precession and procession is the same as that in the 1D case, i.e., the interplay between intrinsic mobility (arising from the feedback inhibition) and the extrinsic mobility (arising from the external input) generates oscillatory tracking in the network. Our simulation result (not shown in the current manuscript) shows that the network bump in the 2D arena sweeps back and forth around the external input along the tangent of the trajectory.
>
> 2) We would love to emphasize that the study of the 1D case is not simple, which matches the setting of most rodent experiments on studying theta phase coding, that is, these experiments were done when animals run on a linear track (a few of them were done on the Z-shaped or W-shaped track). Therefore, the study of the 1D case can link directly to the experimental data.
>
> 3) Experimentally, it is unclear yet what theta weeps exactly look like when an animal runs in the open field, due to the lack of recorded neurons, making it hard to accurately decode the animals’ position (via private communication with the authors of the paper, Wang et al. Science 2020). Some experimental scientists believe that theta sweeps in the 2D space is similar to the 1D case, as predicted by our model. But Edvard I. Moser gave a talk at the Cosyne conference several year ago, indicating that theta sweeps of grid cells in the 2D space may consist of an intermittent switch between left and right sweeps in consecutive theta cycles. It will be very interesting to model theta sweeps in the 2D CANN in a more accurate way when more concrete experimental data are available.
>
> **About the synaptic plasticity:**
>
> It is an interesting issue to explore what kind of synaptic plasticity drives the form of a CANN, as well as the effect of synaptic plasticity during theta phase shift. Indeed, synapses in the hippocampus can be very plastic during the postnatal period or during reward-based learning. However, in the present study, we focused on investigating the mechanism of generating theta phase shift of place cells, a phenomenon widely observed during free locomotion. Therefore, we consider that the CANN has been preconfigured through previous training (see below on how a CANN can be learned through experiences). Notably, the Gaussian form of synaptic connections in our model is chosen for the convenience of theoretical analysis. In practice, our main results remain unchanged when the synaptic connections are not of the idealized form or fluctuated due to continuous plasticity, as long as the synaptic strength distribution follows the basic principle, that is, nearby cells have stronger connections and faraway cells have weaker connections, as well as the feedback inhibition exists (remarkably, in the case of non-idealized synaptic connections, the feedback inhibition, which tends to destabilize the bump state, has a positive contribution to avoid that the network dynamics is trapped in local minimums.).
>
> The issue of “learning a CANN through synaptic plasticity” has been largely studied previously. It has been demonstrated that the synaptic distribution needed by a CANN can be learned through local Hebbian learning or STDP [1]. For example, this study [2] demonstrated that a ring attractor for the head direction system can be learned by a very simple biological plausible learning rule [3]. Also, an early work by Blum & Abbott [4] and a recent work [5] demonstrated that the STDP rule can lead to the translation-invariant synaptic connections.
>
> **On the link between firing rate based CANN model and the precise spike timing of theta phase precession and procession in place cells’ firing:**
>
> In our model, phase precession and procession of a neuron are quantified by the shifts of the timing of peak firing rate of the neuron in consecutive cycles. The similar measurement has been used in modelling studies in the field for many years, see, e.g., the classical dual oscillatory model for explaining phase precession [6]. Turning the firing rate into Poisson spikes in our model is straightforward, which generates the same phase shift phenomenon, although it is a bit noisy. Notably, in the experiments, theta phase precession was displayed by aggregating spikes over many theta cycles (since the timing of individual spikes is very noisy), which is effectively a rate-based plot, similar to Fig.5C in our paper.
>
> We hope our replies have clarified the concerns of the reviewer.

---

> > ### Author Response · Authors · 2022-08-01
> > **References:**
> >
> >
> > **References:**
> >
> > [1] Bi, G. Q., & Poo, M. M. (1998). Synaptic modifications in cultured hippocampal neurons: dependence on spike timing, synaptic strength, and postsynaptic cell type. Journal of neuroscience, 18(24), 10464-10472.
> >
> > [2] Vafidis, P., Owald, D., D'Albis, T., & Kempter, R. (2022). Learning accurate path integration in ring attractor models of the head direction system. Elife, 11, e69841.
> >
> > [3] Urbanczik, R., & Senn, W. (2014). Learning by the dendritic prediction of somatic spiking. Neuron, 81(3), 521-528.
> >
> > [4] Blum, K. I., & Abbott, L. F. (1996). A model of spatial map formation in the hippocampus of the rat. Neural computation, 8(1), 85-93.
> >
> > [5] George, T. M., de Cothi, W., Stachenfeld, K., & Barry, C. (2022). Rapid learning of predictive maps with STDP and theta phase precession. bioRxiv.
> >
> > [6] O'keefe, J., & Burgess, N. (2005). Dual phase and rate coding in hippocampal place cells: theoretical significance and relationship to entorhinal grid cells. Hippocampus, 15(7), 853-866.

---

### Official Review · Reviewer_yCJA · 2022-07-11

**Rating:** 7
**Confidence:** 4
**Soundness:** 4 excellent
**Presentation:** 3 good
**Contribution:** 3 good

**Summary:**

The authors propose an attractor network model which includes a “feedback inhibition” term that jointly exhibits phase precession and procession in hippocampal place cells.  The model is defined via standard neural network rate equations in one-dimension using recurrent connections with a Gaussian profile.  Feedback inhibition to a neuron encoding location x is modeled as a dynamic term that depends only on the activity of that neuron (it is a low pass filter of the synaptic input to neurons encoding location x).  This is described qualitatively as the “overall effect of the interplay between the hippocampus and the medial septum”.  The consequence second order dynamics gives rise to traveling wave solutions.  The competing effects of the external input and intrinsic traveling wave dynamics induces oscillatory behavior in a region of phase space that accounts for the jointly observed precession and procession.  The authors also show how this model can be tuned to account for both “unimodal” and “bimodal” cells.

**Questions:**

I have a couple of presentation related points and one main issue regarding the formulation of the model.

Regarding the model:  Using a term like equation 3 brushes a lot under the rug in terms of how the system itself works.  The second order dynamics that arises seems to exhibit the desired effect, and it is great to have identified a dynamical system that shows the appropriate empirical effects, but this formulation leaves as a mystery exactly how the “interplay between the hippocampus and medial septum” generates this effective inhibition term.   As it is, the term essentially just introduces the simplest second order temporal dynamics into equation (1) that (along with the Gaussian profile weights) turns it into an effective wave equation.  This is all great in terms of identifying the possible mechanism, but it would be nice to have some idea of what model of such interactions would give rise to such a term.  (Note that as I write this, this criticism sounds more severe than I intend it.)

Regarding presentation:  I think lines 217-231 could be greatly improved so the reader has to do less work.  In particular, it could be more immediately clear how h(t) relates to the position relative to the probe neuron (you erase this by defining h(t) as an absolute value).  It was easier for me to think in terms of the actual distance from the probe, personally.  Putting the sign back and considering the separate cases was an easier way to think about it for me.

Finally, the language around “forward/reverse theta sequences” and “forward and backward” sweeps of decoded position is likely very confusing for readers who are not sufficiently familiar with the place cell literature.  (There is some probability that they have confused this reviewer and I have fundamentally misunderstood the paper.).

**Limitations:**

I agree with the authors that there are negligible societal impacts of this work.

**Strengths And Weaknesses:**

This papers presents a simple and elegant model that incorporates multiple observations into a single framework (unimodal and bimodal cells).  It is clearly presented and the figures illustrate well all main conceptual points (the example movies in the supplementary material are also excellent and helpful).   The structure of the model is sufficiently detailed with all important steps of the construction and derivation of results presented.  The paper appears to exhibit overall high quality and clarity (though there are a few places where writing can be cleaned up and polished).

The results also appear original (to me, though I am less familiar with this section of the literature on hippocampal place cells than I used to be) and significant (in that they demonstrate a model that jointly exhibits phase precession and procession - again, to me).

 The primary weaknesses I see are addressed in “Questions” below.

---

> ### Author Response · Authors · 2022-08-01
> **Response to reviewer yCJA:**
>
> **We thank the reviewer for encouraging comments about our paper and valuable suggestions for improving the presentation.**
>
> **About the formulation of the model, i.e., the detailed realization of feedback inhibition through the interplay between the hippocampus (HPC) and the medial septum (MS).**
>
> Thanks for pointing out this issue of biological implementation. The short answer is that we are still not very sure about the biologically detailed implementation of feedback inhibition for theta phase shift. In reality, there are two possible ways to implement such feedback inhibition: one is the spike frequency adaptation (SFA) widely observed in neuronal firing in the brain [1,2], whose effect is to counterbalance neural firing, and the other is the interplay between HPC and MS. The reason for us to consider the interplay between HPC and MS in the present study is that MS acts as a pacemaker to generate hippocampal theta in the experiments: lesions of MS abolished theta oscillations in the hippocampus (and hence theta phase precession) [3], and cooling MS slowed down the theta rhythm [4]. Experimental data also recognized an interaction pathway between HPC and MS [5,6] (as discussed in the paper line.313-323): HPC pyramidal cells (place cells) project to GABAergic neurons in HPC which further projects to GABAergic neurons in MS; in return, GABAergic neurons in MS project back to GABAergic neurons in HPC which inhibit place cells’ activities. Such interaction loop might implement the feedback inhibition effect used in our model. Nevertheless, we would like to point out that the current experimental data do not exclude the possibility of SFA (feedback inhibition at the single neuron level) in generating theta sweeps, and more experimental data are needed for establishing a detailed model for the HPC-MS pathway.
>
> We will include more discussions about the biological implementation of feedback inhibition in the revised paper.
>
> As recognized by the reviewer, we adopted a simple mathematical form of feedback inhibition with the aim for elucidating the possible mechanism of generating theta phase shift clearly. We expect that no matter what kind of detailed mechanism the hippocampus adopts to implement such feedback inhibition, as long as the feedback inhibition exists, the CANN can generate the oscillatory tracking behavior, which leads to theta phase shift.
>
> **Regarding the presentation in line 217-231：**
>
> Thanks very much for the suggestion! We removed the absolute form of h(t) and updated the descriptions in line 217-231 to present the results more clearly (see line.217-232 in the rebuttal pdf).
>
> **About theta sequences and theta sweeps:**
>
> Thanks for the suggestion! We will differentiate them clearly in the revised paper to avoid confusion. Basically, theta sequences refer to the sequential firing of place cells in each theta cycle, while theta sweeps refer to that the decoded position (typically reconstructed based on theta sequences by using a standard Bayesian decoding algorithm) sweeps back and forth around the physical position of the animal. They are the two sides of the same coin.
>
> We hope our replies have clarified the concerns of the reviewer.
>
> References:
>
> [1] Alonso, A., & Klink, R. (1993). Differential electroresponsiveness of stellate and pyramidal-like cells of medial entorhinal cortex layer II. Journal of neurophysiology, 70(1), 128-143.
>
> [2] Ha, G. E., & Cheong, E. (2017). Spike frequency adaptation in neurons of the central nervous system. Experimental neurobiology, 26(4), 179.
>
> [3] Partlo, L. A., and Sainsbury, R. S. (1996). Influence of medial septal and entorhinal cortex lesions on theta activity recorded from the hippocampus and median raphe nucleus. Physiol. Behav. 59, 887–895. doi: 10.1016/0031-9384(95)02208-2
>
> [4] Petersen, P. C., and Buzsáki, G. (2020). Cooling of medial septum reveals theta phase lag coordination of hippocampal cell assemblies. Neuron 107, 731–744.e3. doi: 10.1016/j.neuron.2020.05.023
>
> [5] Toth, K., Borhegyi, Z., & Freund, T. F. (1993). Postsynaptic targets of GABAergic hippocampal neurons in the medial septum-diagonal band of broca complex. Journal of Neuroscience, 13(9), 3712-3724.
>
> [6] King, C., Recce, M., & O’keefe, J. (1998). The rhythmicity of cells of the medial septum/diagonal band of Broca in the awake freely moving rat: relationships with behaviour and hippocampal theta. European Journal of Neuroscience, 10(2), 464-477.

---

### Official Review · Reviewer_kP6x · 2022-07-20

**Rating:** 5
**Confidence:** 3
**Soundness:** 4 excellent
**Presentation:** 3 good
**Contribution:** 3 good

**Summary:**

  The authors of the current submission set out to provide a biologically inspired "neural field" (herein a PDE model) model of the experimental data reported in Wang et al. (2020).
  The proposed model, unlike those previously proposed, accounts for "phase procession" (cf. phase precession), as reported in Wang et al.
  To incorporate the above phenomenon, the usual PDE equations are combined with "feedback inhibition".
  The resulting system is shown to undergo a Hopf bifurcation giving rise to "oscillatory tracking", which is proposed as an explanation for the coexistence of phase precession and procession.
  Additionally, neurons with "unimodal" and "bimodal" tuning functions (as measured wrt to the theta oscillation) are shown by varying model parameters thus giving a unified description of the two.

**Questions:**

Regarding clarity:
1. Having read Wang et al. I found that their subfigure 1A (which is also reproduced in the appendix of the current submission Figure SI 1 ) to be a clearer explanation of phase precession vs procession. I believe a schematic in the vein of SI 1B would make for an easier explanation than the current Fig. 1B from the main text.

2. There is little information as to what the heatmap in Fig 5C represents, and a fortiori how this normalized histogram was computed.

Terminology:
The authors say the model is a "continuous attractor neural network". Please correct me if I am wrong, but the incorporation of feedback inhibition intentionally de-stabilizes the bump attractor; it appears that feedback inhibition was *specifically* added to allow for a Hopf bifurcation.
If I understood it correctly, I believe this ought to be clarified that though derived from a CANN, the model is not a CANN.

Misc:
$I^{ext}$  is modeled assumed to be linear function of time. I would find it interesting to compare numerically how the proposed model deals with stochastic $I^{ext}$ in comparison with previously proposed models.
Neutrally stable dynamics in conjunction with a white noise drive ought to give rise to a diffusing receptive field. I believe the proposed model will not be susceptible to such behavior.


**Strengths And Weaknesses:**

Admittedly I was not familiar with Wang et al. (2020), which attests to my limited ability to judge the significance of this submission.
That being said, I have since familiarized myself with the works citing Wang et al and I believe the contribution offered by the current submission to be novel.
Overall, I found the writing and derivations to be very clear. Passages and figures requiring further are mentioned in the `Questions` field below.

In sum, I believe this to be a technically sound, interesting contribution.
Nevertheless, I hesitate to enthusiastically recommend it for acceptance for the following reason:

1. As stated I cannot accurately gauge the significance of this submission, especially to the NeurIPS community.
2. Because the submission is devoted to deriving a particular model of place cell function (ie. without relying on statistical or machine learning methods, nor without any immediate implications for those communities), it might be an equally good if not better fit in specialized journals like PlOS CB, Neural Computation, eLife etc.
Naturally, this is no grounds for dismissal but I would like to allow fellow reviewers and the authors to discuss whether NeurIPS is the appropriate venue for this work.

---

> ### Author Response · Authors · 2022-08-01
> **Response to reviewer kP6x:**
>
> **We thank the reviewer for encouraging comments and valuable suggestions for improving the presentation.**
>
> According to the reviewer’s suggestions, 1) we have replaced Fig.1B with a new figure to better illustrate theta phase precession and procession (please see the new Figure 1 in our rebuttal pdf); 2) we will add more descriptions to explain the plot of Fig.5C in the revised paper. Basically, Fig.5C follows the same form of Fig.3B in the experimental paper (Wang et. al, Science 2020), where the authors aligned phase shifts of neurons according to the normalized position of the animal in the place field of each neuron and then calculated the probability of phase shift (the detailed calculation process is introduced in Sec 4.2 in SI).
>
> **About publishing our work at NeurIPS:**
>
> **We strongly believe that NeurIPS is the right venue to publish our work. The reasons are below:**
>
> 1) Since it was founded in 1987, NeurIPS (formerly NIPS) have accepted computational neuroscience papers. Especially in the early years, it was mainly targeted for neuroscience modeling. Nowadays, even though it flooded with deep learning papers, it still accepts many papers on neuroscience and brain per year, e.g., about 140 submissions in 2018 and 170 in 2019. We ourselves have been publishing and reviewing computational neuroscience papers in NeurIPS for many years.
>
> 2) Closely related to our study on the topic of hippocampal/entorhinal modeling, here are some papers published recently at NeurIPS:
>
> a)	Whittington, J., Muller, T., Mark, S., Barry, C., & Behrens, T. (2018). Generalisation of structural knowledge in the hippocampal-entorhinal system. Advances in neural information processing systems, 31.
>
> b)	Sorscher, B., Mel, G., Ganguli, S., & Ocko, S. (2019). A unified theory for the origin of grid cells through the lens of pattern formation. Advances in neural information processing systems, 32.
>
> c)	Vértes, E., & Sahani, M. (2019). A neurally plausible model learns successor representations in partially observable environments. Advances in Neural Information Processing Systems, 32.
>
> d)	Gao, R., Xie, J., Wei, X. X., Zhu, S. C., & Wu, Y. N. (2021). On Path Integration of Grid Cells: Group Representation and Isotropic Scaling. Advances in Neural Information Processing Systems, 34, 28623-28635.
>
> e)	Nayebi, A., Attinger, A., Campbell, M., Hardcastle, K., Low, I., Mallory, C. S., ... & Yamins, D. (2021). Explaining heterogeneity in medial entorhinal cortex with task-driven neural networks. Advances in Neural Information Processing Systems, 34, 12167-12179.
>
> 3) Although the current study mainly focuses on modeling the firing properties of place cells, it has strong implications to learning algorithms in AI. As we stated in Line.34-38, theta sequences of place cells essentially compress the temporal structure of animal experiences from a behavior time scale (seconds) to a time scale compatible with the STDP learning rule (<100 ms), which enables the learning in neural systems. Our model unveils the underlying mechanism of implementing this compression. This may inspire us to develop biologically plausible learning algorithms in AI to process similar learning tasks, e.g., navigation in the real world, planning and decision making.
>
> **CANN v.s. Hopf bifurcation：**
>
> We agree with the reviewer about the terminology concern. Indeed, strictly speaking when the negative feedback is relatively strong, the network is no longer a “CANN”, since the bump is no longer stationary in the attractor space. However, we prefer to keep using the term CANN through the paper (as this is conventionally used in the neuroscience society) to avoid confusing readers with only neuroscience background. We will, however, clarify this point in the revised paper.
>
> **On the stochasticity of I^{ext}：**
>
> If only a white noise input (setting I^{ext}=0) is applied to the network, the bump will experience diffusive dynamics (as the reviewer mentioned) or super-diffusive dynamics (depending on the strength of negative feedback). If white noises are added on top of the external input, it will not change our main results. The external input I^{ext} in our model mimics the physical location of the animal during locomotion which usually continuously changes over time. Including white noises in I^{ext} will induce small fluctuations of the animal location, but the oscillatory tracking behavior of the network remains unchanged.
>
> We hope our replies have clarified the concerns of the reviewer.

---

### Author Response · Authors · 2022-08-09
**General reply to all reviewers -- any feedback?**

Dear All:

We appreciate the reviewers for their questions/comments and their time and effort in reviewing the paper. Since we haven't got any feedback from the reviewers (the deadline is Aug 09 '22 08:00 PM UTC), we are wondering if the updates to the manuscript and replies to the corresponding questions resolve the raised concerns. If this is indeed the case, we hope the reviewer could consider updating the score accordingly. If not, we hope the reviewer could clarify any remaining concerns/questions and we are happy to engage in further discussion.
We again thank the reviewers for the thoughtful comments and for the time and effort in reviewing the paper.

Yours sincerely,

Paper3363 Authors

---

### Meta-Review · Area_Chair_PEKS · 2022-08-24

**Recommendation:** Accept
**Confidence:** Less certain

**Metareview:**

This paper presents non-trivial and novel theoretical and computational modeling that accounts for experimentally observed phenomena: the theta phase procession and precession. These phenomena are implicated in the neural representation and learning of neuronal networks involving hippocampus. Although the current manuscript does not address the learning and the presentations are limited to a linear track environment, it represents a clear advance in linking spatial and temporal information representations by extending the standard continuous attractor models that do not exhibit phase procession/precession. Furthermore, the model elegantly explains the biologically observed unimodal and bimodal cells. The authors are encouraged to improve the clarity of some parts of the writing.


**Award:**

No

---

### Decision · Program_Chairs · 2022-09-14

Accept